# Optimization Framework for Early Conceptual Design of Helicopters

Ludvig Knöös Franzén [1,*], Ingo Staack [1], Petter Krus [1] and Kristian Amadori [2]

1    Division of Fluid and Mechatronic Systems, Linköping University, 58183 Linköping, Sweden
2    Overall Design and Survivability, Saab Aeronautics, 58254 Linköping, Sweden
*    Correspondence: ludvig.knoos.franzen@liu.se; Tel.: +46-13-284079

**Abstract:** This work illustrates how a proposed method can be used to create optimization frameworks for early conceptual design studies and to increase overall knowledge at an early design stage. The method is intended to facilitate concept selection in challenging domains that typically involve multidisciplinary design problems with contradictory requirements. The main focus of the work presented here is on the conceptual design of helicopters; however, the method is intended to be applicable to early design studies in other domains as well. In short, statistics about existing helicopters are collected and compiled to provide a basis for various regression analyses. The purpose of this is to unravel relationships in the data and to obtain simple estimation models from statistical regressions that can be used in conjunction with existing formulas and equations to generate an initial helicopter design estimate. Models for each discipline, such as aerodynamics, are then created using the outcomes of the regression analyses and existing equations. Lastly, the method is used to define a multidisciplinary design optimization framework incorporating all the models obtained from the different disciplines. A case study based on search and rescue operations is used to test the proposed framework in order to obtain possible first suggestions for the baseline design of a new general-purpose search and rescue helicopter.

**Keywords:** symbolic regression; optimization; conceptual design; genetic algorithm; design of experiments; helicopter statistics

## 1. Introduction

Early design studies are intended to increase overall knowledge about new concepts and to indicate possible candidates for further evaluation. However, generating and deciding between a large number of suitable concepts can be a challenging task, especially when considering the multiple design disciplines involved. Helicopter design is such an example and is a multidisciplinary domain that often includes contradictory requirements and negotiations to arrive at suitable concepts. For example, objectives such as a desired number of people onboard is typically in conflict with specified weight requirements. Overall production cost is another factor that often places additional constraints on a helicopter design, as shown in [1]. Consequently, optimization becomes a powerful tool that can alleviate the designers' work and facilitate decision-making. Multidisciplinary design optimizations (MDOs) are a commonly used approach in aerospace design problems and involves more considerations than just optimizations. As an engineering discipline, MDO additionally deals with topics such as sensitivity analyses, information processing, and management strategies [2]. It can also be seen as a methodology where the interactions between involved disciplines and subsystems are exploited and in focus for the design of new systems. However, these interactions introduce additional challenges compared to optimizations in a single discipline. According to [3], the two main challenges that an MDO approach brings are in terms of computational expense and organizational complexity. Optimizations and MDOs have been used in several helicopter design studies. For example,

the work presented in [4] shows a study where a rotorcraft multidisciplinary design and optimization framework was utilized to assess regenerative powerplant configurations for preliminary design. Another study shows how a multidisciplinary framework can be used to perform both single and multiobjective optimizations in order to reduce fuel consumption and emissions for new rotorcraft designs [5]. The obtained results were here subsequently used to explore the design space for operations at a mission level and to see how varying flight parameters influenced the objectives. The work from [1], that was mentioned previously, illustrates how the production cost for a light turbine-training helicopter can be minimized together with several other optimization objectives in an MDO framework. The framework also enables explorations of the results so that designers can see the influence of the design variables for all involved disciplines simultaneously. High-fidelity rotorcraft computations are performed in [6] by means of a multidisciplinary design and optimization framework. The involved disciplinary models include computational fluid dynamic codes and have, for example, been used with optimizations to reduce rotor torque while maintaining other desired rotor blade parameters, such as rolling and pitching moments.

As just seen, the early design of a helicopter involves many considerations, such as the intended mission, range and required power. While many different designs may meet the intended requirements, it is not guaranteed that they will do so in an optimal way. It is therefore important to properly explore the available design space at the conceptual stage in order to identify the most suitable solutions that can meet the overarching requirements and needs in the best way. However, consideration of all these details makes the design process more complex, and finding a "best solution" without optimization could be both time-consuming and costly. Furthermore, the limited knowledge about a new concept that is available at the beginning of a design process further adds to the uncertainty. It is therefore desirable to increase overall knowledge about a new design as much as possible early on in the process. There are often well-established methods or general approaches and guidelines for conceptual design studies in different engineering domains, such as the ones presented in [7–10] for helicopter design or by Raymer for aircraft design [11]. However, these, and some of the frameworks mentioned before, are typically fairly detailed, and a designer must be quite familiar with the domain and process in question to make an initial concept selection. Consequently, it is desirable to facilitate an early concept selection through other and simpler means. One way of achieving this is to estimate the characteristics of new concepts using statistics drawn from existing solutions. Such "ballpark" estimates are usually enough at the beginning of a design process and in the evaluation of possible initial concepts. Sensitivity analyses can then indicate how sensitive a concept is to changes in various parameters and may thus also give an indication of the required model fidelity. More details are, of course, preferable to a certain extent. However, if a concept is fairly insensitive to changes in some of the involved models, statistical estimations are probably sufficient for achieving good initial values. Requirements might also change over time, and consequently, it is important to be able to trade requirements and investigate possible future scenarios so that resilient designs can be obtained.

This work therefore proposes a method for generating optimization frameworks for early conceptual design studies. In the case of this paper, the method is used to generate an MDO framework for the conceptual design of a helicopter in order to address the challenges mentioned above. This helicopter optimization framework is mainly built from statistical regression models that were obtained in different ways. Together, these models make up the MDO framework, which is implemented in the modeFRONTIER software [12]. From there, the constraints and objective functions can be adjusted to meet the designer's specific design task. Design and trade space explorations can thereafter be performed in order, for example, to investigate the concept's sensitivity to the underlying models. In this work, the design of a general-purpose search and rescue (SAR) helicopter is used as a case study to test the implementation of the framework and to obtain a Pareto front of possible solutions. Consequently, the SAR objective functions of the case study describe a helicopter that

can fly as far as possible with as many passengers as possible, while minimizing weight, fuel consumption, and the average cost per flight hour. Constraints, such as a maximum flight velocity, are introduced to keep the solutions within realistic limits. Proposals and opportunities for future work on the method and optimization framework are presented at the end of the paper, together with a brief discussion on the outcome of the case study.

## 2. Approaches and Design Methodologies

This section briefly explains some of the governing equations for helicopter design as well as approaches that can be used to create statistical regression models for optimizations in order to meet the intended optimization framework outlined in the introduction. Additionally, at the end of this section, other initiatives that have utilized optimizations for helicopter conceptual design are identified.

### 2.1. Helicopter Equations and Relationships

An essential formula in any air vehicle design is the Breguet range Equation (1). This equation can be used to calculate the distance that an aircraft can fly given a set of parameters in steady and level flight [13].

$$Range = \frac{V}{g} \frac{1}{SFC} \frac{L}{D} ln\left(\frac{W_{initial}}{W_{final}}\right) \tag{1}$$

Here, $V$ corresponds to the aircraft cruise velocity, $g$ = the gravitational constant, $SFC$ = specific fuel consumption, $L$ = lift force, and $D$ = drag force, while $W_{initial}$ and $W_{final}$ describe the ratio between the weight at the start and the end of a flight. As mentioned previously, this formula can be used to estimate the range of air vehicles with surprisingly good results [14]. However, some additional factors must be taken into consideration when estimating the range for helicopters. It can be seen in [10] that two additional parameters are added to the Breguet range equation for helicopters, namely, a rotor efficiency factor, $\eta$, and a coefficient, $\xi$, that takes power losses in the transmission into account. This "updated" range equation for helicopters is shown in Equation (2):

$$Range = \frac{V}{g} \frac{\eta\xi}{SFC} \frac{L}{D} ln\left(\frac{W_{initial}}{W_{final}}\right) \tag{2}$$

The $\frac{L}{D}$ term in Equation (2) can be estimated using a relationship described in [15] that can also be seen in Equation (3) below.

$$L/D = WV_\infty/P \tag{3}$$

In this equation, $V$ is once more the velocity of the helicopter. $W$ is the weight of the airborne helicopter and $P$ is the available main rotor power. The flight velocity and main rotor power are correlated in a way that can be described using a graph presented in [16]. A representation of this relationship can be seen in Figure 1.

There are many more proposed formulas that can be found in the literature. However, finding simple equations for helicopter design, without entering into too much detail, is difficult. Regression models based on statistical data for already existing solutions can therefore be a good complement to the calculations at a conceptual stage. Additionally, many of the equations used in helicopter and airplane design are originally based on statistics for existing or no-longer-operating aircraft.

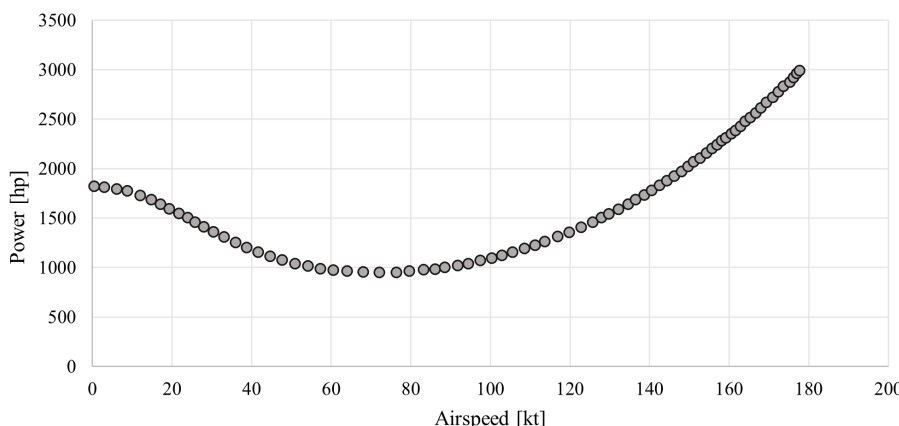

**Figure 1.** The relationship between helicopter power and airspeed. Based on the graph presented in [16].

*2.2. Statistical Regression*

A statistical regression can be described as a method for determining dependencies between variables of interest in a dataset. One of the simplest forms of regression is trendline fitting on graphs between two variables [17]. In this case, the trend line can give the relationship between two parameters; for example, as a linear equation. Different equations may fit the investigated data better, such as an exponential trend line. However, it can be difficult to judge whether one trend line fits the data better than another. A possible way of assessing the fit is to use, for example, an $R^2$ value. The $R^2$ value, also called the coefficient of determination, simply describes how well a regression model fits the data on which it is based [18]. The $R^2$ value is built on the variance of the dataset and can be described by the following Equation (4):

$$R^2 = \frac{Variance\ explained\ by\ the\ model}{Total\ variance} \tag{4}$$

The $R^2$ value in this case is given as a number in the range 0–100%, where 100% would correspond to a perfect fit. However, having a high $R^2$ value and a perfect fit may not necessarily be either good or desirable. For example, both interpolations and extrapolations using the trend line's equation can give values that are very wrong, and consequently, some intuition is almost always necessary when fitting a trend line. Hence, this type of regression can be prone to biases [17].

Regression analyses with more than two variables are commonly referred to as multiple regression analyses [19]. These analyses typically use two or more independent variables with known values to estimate the value of a single dependent parameter. Unlike a "regular" regression with only one variable, multiple regression analyses include weightings so that the contribution of each predictor variable can be determined.

Both regular and multiple regression analyses can be performed on the logarithm of the original values in a dataset, and there are many potential benefits of doing this. One example is that the data can be better "centred", which can allow for a better fit with a linear regression model. It thus also facilitates the analysis of datasets that are not normally distributed [20].

Symbolic Regression

Fitting suitable trend lines and creating statistical models under uncertainty can be challenging. However, optimization can be used for this purpose. One way of doing so is to use symbolic regression. This, as the name implies, is a regression analysis that yields prediction models using mathematical expressions that best fit a given dataset. Lengthy mathematical expressions are typically penalized so that both accuracy and simplicity are considered [21]. Optimizations using, for example, genetic algorithms are thus

suitable because the mathematical building blocks—such as addition, subtraction, and multiplication—can be seen as the different available chromosomes. An optimization is then performed with the objective of minimizing aspects such as the mean square error, while penalizing the complexity of the resulting expressions or individuals. This reduces some of the human bias and creates a "simple" but accurate prediction model from statistical data using optimizations. The end result of a symbolic regression using genetic algorithms is therefore an equation, or formula, optimized to fit the given data while keeping the expression simple in terms of length and complexity. Figure 2 provides an example of a symbolic regression process.

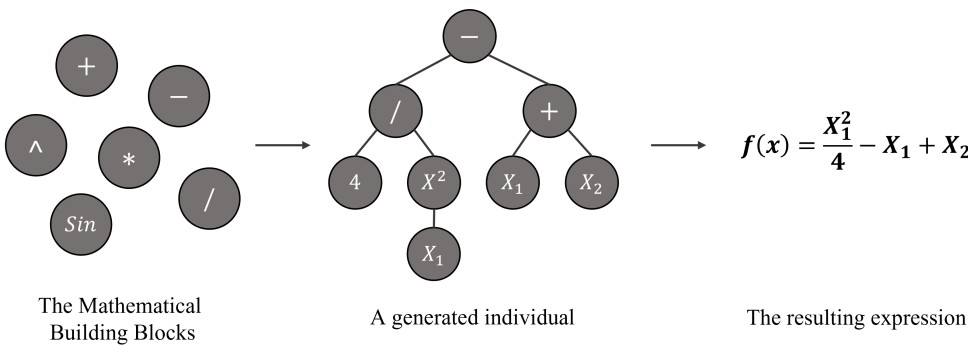

**Figure 2.** An illustration of a symbolic regression process.

### 2.3. Optimization

So far in this paper, genetic algorithms have been suggested as one optimization technique. However, there are many other methods, ranging from simple gradient-based approaches to more complex particle swarm optimizations [11,22]. Common to all optimization techniques is the fact that there typically exist objective functions, constraints, and design variables. Optimizations with more than one objective function are referred to as multiobjective optimizations (MOOs) and optimizations within more than one discipline, such as aerodynamics, are labeled as multidisciplinary design optimizations (MDOs) [23,24]. Optimizations, and especially MDOs, can be quite expensive in terms of computational cost and time. It is therefore sometimes beneficial to create surrogate or meta models that can be used to make predictions about the optimization results. However, in order to create accurate surrogate models, certain benchmark optimizations must be performed. These optimizations should be chosen so that as few simulations as possible need to be performed in order to create an accurate surrogate model. A design of experiments (DOE) can be used to determine the simulations that should be performed in order to obtain suitable elements of the surrogate model [25]. A variety of different DOE methods can be used. One example of a common DOE is Latin hypercube sampling.

Optimization algorithms are available and can be found in many different software and programming languages, such as MATLAB, Microsoft Excel, Python, etc. The software called modeFRONTIER [12] is dedicated to design optimization and allows users to perform design space explorations on the optimization results in order to identify optimal solutions in an intuitive way. Additionally, modeFRONTIER allows for MDO with models from other tools, and it features several optimization algorithms and design space sampling methods.

### 2.4. Conceptual Helicopter Design Initiatives

Conceptual design studies of aircraft are usually performed based on historical data of already existing solutions. A well-known methodology for aircraft conceptual design, which utilizes both statistical data and some optimizations, is presented in Raymer's book *"Aircraft Design: A Conceptual Approach"* [11]. However, this book mainly focuses on fixed-wing aircraft and does not specifically address the conceptual design of rotorcraft or helicopters. Although very different, some inspiration can still be taken from the conceptual design methodology for fixed-wing aircraft and applied to rotorcraft in early design studies.

This can, for example, include the overall sizing procedure of a cabin or how different weight penalties must be taken into account for engine installations, etc. Besides the helicopter design references mentioned in the introductory section, the *"Helicopter Flying Handbook"* provides a detailed overview of all aspects of rotorcraft [26]. From a conceptual design perspective, the handbook can, for example, give guidance about different configuration possibilities and the desired flight characteristics of a concept to be developed and evaluated. However, there are various dedicated guides and methodologies available for the conceptual design of helicopters and rotorcraft. One of these is the *"Guide for Conceptual Helicopter Design"* by Kee [7]. Here, a combination of historical data and formulas are used for the conceptual design of a single-rotor utility helicopter. More recent initiatives for the conceptual design of rotorcraft have been presented, for example, in studies by the German Aerospace Center (DLR) [27] and the Georgia Institute of Technology [28]. A multidisciplinary design framework for rotorcraft is introduced in [27]. This framework also utilizes optimization and a high level of modularity for achieving flexibility in the process and among the different disciplines. The study in [28] introduces a conceptual design methodology for rotorcraft maneuverability using modeling and simulation. Design space explorations and optimizations are also used here to give decision support and to indicate a suitable design region in terms of maneuverability.

## 3. Method

This chapter presents the overall method that is used to create the helicopter conceptual design optimization framework suggested in this paper. The method builds upon the gathered knowledge from Sections 1 and 2, but is also intended to be general so that the design of any system is facilitated. Figure 3 presents an illustration of the method and intended workflow to create an optimization framework for early design studies in general.

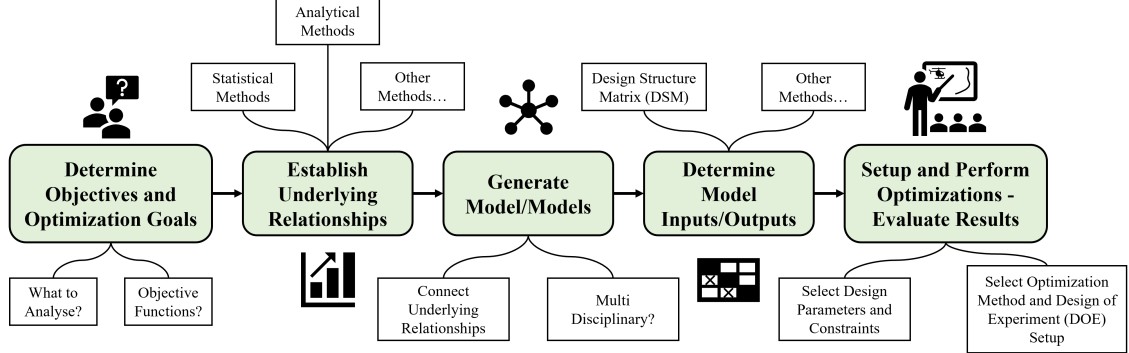

**Figure 3.** The general method and workflow for creating optimization frameworks for early design studies.

As seen in Figure 3, the process starts by determining the purpose of the analysis. For example, are the design studies going to evaluate helicopters, cars, or toasters? Once the topic of analysis is decided, objective functions can be specified. For example, this could consist of the main objectives or requirements that the design is intended to fulfill, such as the capability of transporting a given number of passengers over a specific range in as little time as possible. With goals and objectives determined, the analysis of underlying relationships can begin. This step typically involves applying statistical data from previous solutions to the objectives and goals specified in the previous step. This could, for example, consist of various regression analyses, such as the ones described in Section 2.2. It is also possible to utilize analytical methods, such as centre-of-gravity calculations for the involved components. Other methods, such as domain-specific or well-established formulas, can, of course, be included here as well; for example, Newton's second law. The purpose of this step is essentially to identify the relationships between the optimization objectives and possible design variables. These relationships can then be used to establish different

calculation models related to the chosen objectives. As the *Generate Model/Models* step in Figure 3 shows, these models can either be single large models or split into several smaller models—for example, one for each involved discipline—that can be connected to each other.

Once appropriate models have been created, the inputs and outputs between each model must be determined. In an MDO case, a design structure matrix (DSM) can aid in this specific task [29]. A DSM gives an overview of the connections between the different involved models and can be used, among other things, to sort the execution order of these models in the subsequent optimization. Finally, appropriate design parameters and constraints can be added to the optimization, depending on the previously determined objectives and goals. Once this is complete, the optimization itself can be set up by choosing a desired optimization method and, for example, sampling the design space using a DOE. The optimization results can subsequently be investigated and evaluated, as well as being used to aid in decision support, design, and trade space explorations. The proposed workflow presented in Figure 3 can be somewhat iterative, and certain steps might need to be revisited throughout the process as more knowledge about a design is generated. However, the final outcome of the workflow is an optimal design point or a Pareto front of overall optimal solutions for the given objectives.

## 4. Case Study and Implementation

A case study for the conceptual design of a general-purpose search and rescue (SAR) helicopter is performed and implemented in order to test the proposed method and workflow illustrated in Figure 3 in the previous section. The first step is to determine the goals and objectives of the optimization that is to be performed. SAR operations is an area that typically involves several goals and objectives that are sometimes in conflict with each other. Generally, the goal is to find a lost rescue subject in as little time as possible while also being able to cover as large a search area as possible. Other considerations, such as fuel consumption for the involved vehicles and rescue/passenger capacity, must also be taken into account. Based on this goal, the objectives for the optimization of a possible new SAR helicopter in this particular case study are listed as follows:

- **Maximize** range and rescue/passenger capacity.
- **Minimize** fuel consumption, weight, and cost per hour.

With the objectives established, the process can proceed to the step of determining the underlying relationships and the definition of suitable models to be used for the optimization. However, some of the design parameters and constraints are already determined and can be introduced at this point for the SAR helicopter case study. In the design of other systems, knowledge might be limited at an early stage. The design parameters and constraints in such a case can typically be decided at a later stage, as indicated in the process presented in Figure 3. The introduced design parameters are here defined as:

- Fuselage length ($\approx$ helicopter size).
- Helicopter rotor diameter.
- Number of rotors.

A constraint on the flight velocity of the helicopter is also introduced at this point in order to avoid problems with stalling rotor blades due to aerodynamic effects at high rotational speeds [26]. The next step in the method is to determine the relationships and formulas that are going to be used in the optimization. The range equation for helicopters (2) is considered to be a central part of this case study because it involves consideration from several disciplines, such as aerodynamic and structural analyses. Consequently, additional relationships had to be found in order to create a model for estimating range. The $L/D$ term in Equation (2) was estimated using the expression stated in Equation (3). The relationship between power and airspeed was estimated using the graph shown in Figure 1. The coefficient of power loss, $\xi$, was specified as 0.88 for a single rotor helicopter in forward flight, while a dual-rotor counterpart was given a value of 0.93 according to [10]. The

reason for the higher power loss in a single rotor helicopter comes from the transmission losses due to a required tail rotor. Finally, the rotor efficiency, $\eta$, was chosen as 0.87 for high flying speeds [10].

Some of the required parameters for the range estimation were hard to find without entering into more details than were required at this early stage. Therefore, the remaining relationships were determined based on statistics from other existing helicopters, as explained in the next section.

### 4.1. Helicopter Statistics and Regressions

In order to generate estimation models from statistics and regression analyses, data for existing helicopters had to be collected and compiled. The majority of these data were collected from specifications publicly available on either manufacturers' data sheets or web pages such as Wikipedia. Specifications for 75 different helicopters were collected and compiled for this case study using Microsoft Excel. The type of helicopters mainly focused on those used in SAR operations; however, some different configurations were included as well, for example, military helicopters. Overall, the specifications collected for the different helicopters were based on those needed for the range estimation, the overall optimization objectives, and the design parameters. Table 1 gives a summary of the different specifications that were collected and used for the regression analyses, together with an example helicopter. The complete dataset used in this paper is available from [30].

**Table 1.** The different specifications that were collected. An AW139 helicopter is included as a reference to show its values and the associated units.

| Specification | Value | Unit |
|---|---|---|
| Helicopter Type | AW139 | |
| No. Crew | 2 | - |
| Passenger Capacity | 15 | - |
| No. People Onboard | 17 | - |
| Length | 16.66 | m |
| Width | 2.26 | m |
| Height | 4.98 | m |
| Helicopter Volume | 187.5 | $m^3$ |
| Empty Weight | 3621.93 | kg |
| Max Takeoff Weight | 6400 | kg |
| Fuel Capacity | 1568 | l |
| SFC | 298 | g/(kWh) |
| No. Engines | 2 | - |
| Power (one engine) | 1142 | kW |
| Total Power | 2284 | kW |
| No. Blades | 5 | - |
| Main Rotor Diameter | 13.8 | m |
| Main Rotor Area | 149.56 | $m^2$ |
| Total Rotor Area | 149.56 | $m^2$ |
| No. Rotors | 1 | - |
| Disk Loading | 42.79 | $kg/m^2$ |
| Cruise Speed | 306 | km/h |
| Never Exceed Speed | 310 | km/h |
| Range | 1061 | km |
| Endurance | 313 | min |
| Service Ceiling | 6096 | m |
| Rate of Climb | 10.9 | m/s |
| Cost per Hour | 4000 | USD |

With these compiled helicopter data, the regression models could be generated. This was mainly achieved using linear regressions based on the logarithm of the original data values. For example, the helicopter length values were investigated against the number of people onboard. The length was then related to both the width and height of the helicopter

using another regression analysis. The outcome of this was an estimated helicopter volume that could subsequently be used in yet another regression for aspects such as the helicopter's structural weight. This procedure was followed until all the required regression models had been established. Figure 4 presents an example for some of the regressions performed on both the logarithms of original values and default ones.

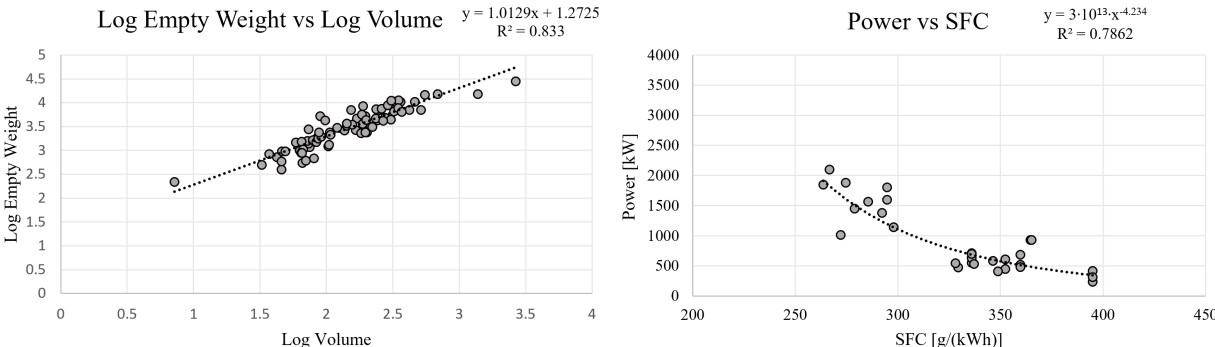

**Figure 4.** A regression for the relationship between helicopter volume and empty weight on a logarithmic scale (**left**). The relationship between specific fuel consumption (SFC) and power (**right**).

As seen in Figure 4, the applied trend lines give both an equation and an $R^2$ value. The $R^2$ values give an indication of how well the trend lines match the underlying statistical values, as mentioned previously in Section 2.2. Even though the $R^2$ values could be improved, the overall trends show good agreement with the underlying data, which indicates that interpolation, as well as light extrapolation, can be performed with some confidence.

Symbolic regressions using genetic algorithms, as mentioned in Section 2.2, were performed on some of the regression analyses where proper relationships and trend lines were difficult to determine "manually". A test to compare the regular and symbolic regression analyses was performed on the relationship between helicopter empty weight and maximum takeoff weight (MTOW). In this case, the regular regression was performed using a linear trend line in Excel. This analysis resulted in the expression shown in Equation (5), with an average error of 18% from the data used.

$$MTOW = 1.8923 \cdot Empty\ Weight - 604.43 \tag{5}$$

The symbolic regression analysis was performed in MATLAB using the GPTIPS toolbox [31]. Simple mathematical building blocks—such as addition, subtraction, multiplication, and division—were added as design parameters for the regression. An optimization was then performed using a genetic algorithm, with the objective of minimizing the mean square error while penalizing lengthy expressions. The optimization setup was specified as a population size of 200 and to run the optimization for 200 generations. Additionally, a tournament method was used with some mutation operators allowed. This particular symbolic regression analysis with a genetic algorithm resulted in the following expression, shown in Equation (6) below:

$$MTOW = 0.00001855 \cdot Empty\ Weight^2 + 1.462 \cdot Empty\ Weight + 504.7 \tag{6}$$

The expression in Equation (6) gave an average error of 12% when compared to the underlying data. Consequently, Equation (6) was able to estimate the MTOW of the utilized dataset 6% better than the regular, manually applied, linear regression in Excel. It should, however, be noted that Equation (6) is a second-order polynomial function with a low gradient. Consequently, heavy extrapolation could result in unreliable results. Nevertheless, all the symbolic regressions performed with genetic algorithms were set up with the same design parameters, objectives, and population specifications as described above.

### 4.2. Optimization Framework

The overall outcome of the regression analyses performed, and the identified helicopter relationships from Section 2.1, was six different estimation models for various disciplines in helicopter design. The established models and their corresponding implementation software are listed below:

- Helicopter Dimensions Model (Matlab).
- Weight Estimation Model (Excel).
- Aerodynamics Model (Matlab).
- Propulsion Model (Matlab).
- Range Model (Excel).
- Cost Model (Excel).

ModeFRONTIER was chosen as a suitable software to connect these different models in a multidisciplinary design optimization (MDO) framework [12]. However, the input and output relationships between the different models had to be established prior to that. As explained above, the helicopter range equation was considered central for this case study. Consequently, all models except the cost model eventually led to inputs for the range model. Table 2 presents an input/output matrix for the six models obtained, together with the intended optimization design parameters, objectives, and constraints.

**Table 2.** The input/output matrix, where inputs are indicated in green and outputs in red. Additionally, design parameters are highlighted in green, optimization objectives in red, and constraints are in yellow.

| Model | Fuselage Length | Rotor Diameter | Number of Rotors | Helicopter Volume | Passenger Capacity | Maximum Takeoff Weight | Fuel Weight | Velocity | Power | Lift to Drag Ratio | Disk Loading | SFC | Range | Cost per Hour |
|---|---|---|---|---|---|---|---|---|---|---|---|---|---|---|
| Helicopter Dimensions | input | | | output | output | | | | | | | | | |
| Weight Estimation | | | | input | input | output | output | | | | | | | |
| Aerodynamics | | input | input | | | input | | output | output | output | output | | | |
| Propulsion | | | | | | | | | input | | | output | | |
| Range | | | input | | | input | input | | | input | | input | output | |
| Cost | | | | | | | | | | | input | | | output |

### 4.2.1. Helicopter Dimensions Model

As seen in Table 2, the helicopter dimensions model takes the fuselage length as an input. The length is then used in different linear regression models to obtain widths and heights that subsequently are used to obtain an estimated helicopter volume. From there, the passenger capacity is derived based on a regression with the previously established helicopter volume.

### 4.2.2. Weight Estimation Model

The weight estimation model uses the previously determined helicopter volume and passenger capacity as inputs. Here, the volume is used to estimate the empty weight with the regression shown in Figure 4. The relationships from the symbolic regression in Equation (6) are then used to derive the MTOW from the estimated empty weight. The

passenger capacity is used to determine the overall payload weight. An average weight of 90 kg is assumed per passenger onboard. MTOW is, in this case, composed of empty weight, payload weight, and fuel weight. Consequently, fuel weight is calculated by subtracting the empty and payload weights from the MTOW.

### 4.2.3. Aerodynamics Model

The rotor diameter, number of rotors, and MTOW are the required inputs for the aerodynamics model. The total rotor area is calculated from here and used with MTOW to determine the helicopter disk loading. A symbolic regression relationship is thereafter used to determine the required power from the disk loading. The relationship from Figure 1 can consequently be used to approximate the flight velocity. Equation (1) is used at the end to calculate the lift to drag ratio, or L/D.

### 4.2.4. Propulsion, Range, and Cost Models

The propulsion model uses the second relationship from Figure 4 and consequently determines the helicopter's specific fuel consumption (SFC) from the input power, as seen in Table 2.

The range model is simply an implementation of the helicopter range Equation (2) from Section 2.1 and requires inputs from most other models. The number of rotors is also added as an input for specifying the value of the coefficient of power loss.

Finally, the cost model determines the cost per hour based on disk loading, as seen in Table 2. This relationship is based on a linear regression between disk loading and cost per hour in log scale.

### 4.2.5. Implementation

With the input and output relationships established in Table 2, a design structure matrix (DSM) could be defined. The resulting DSM with model input and output relationships can be seen in Table 3.

**Table 3.** The design structure matrix (DSM), indicating model inputs and outputs with "X" according to the legend highlighted in blue.

| | Helicopter Dimensions | Weight Estimation | Aerodynamics | Propulsion | Range | Cost |
|---|---|---|---|---|---|---|
| **Helicopter Dimensions** | | | | | | |
| **Weight Estimation** | X | | | | | |
| **Aerodynamics** | | X | | | | |
| **Propulsion** | | | X | | | |
| **Range** | | X | X | X | | |
| **Cost** | | | X | | | |

Legend: Output feedback / (forward) Inputs / Output forward / Feedback Inputs

As seen in both Tables 2 and 3, the required relationships between inputs, outputs, design parameters, optimization objectives, and constraints were determined, and the implementation of the optimization framework in modeFRONTIER could begin. Here, the six different models were connected with each other according to the connections shown in Tables 2 and 3. Additional constraints were also added; for example, to prevent negative values because this was not accounted for in the individual models. The finalized optimization framework generated in modeFRONTIER is given in Figure 5.

The three design parameters in Figure 5 were specified with range properties to indicate the allowed interval of values. The fuselage, or helicopter, length was defined as being able to vary between 4 and 40 m. Similarly, the rotor diameter was allowed to vary between 4 and 25 m. Finally, the number of rotors was defined as either one or two, corresponding to a single- or dual-rotor configuration. The model execution order follows

the structure from Tables 2 and 3, since no "feedback" loops were detected between the models in the DSM. As mentioned previously, additional constraints were added to prevent the fuel weight and lift to drag ratio from taking on too small or negative values. The originally intended velocity constraint was specified as an upper limit of 350 km/h. The objectives of the optimization were finally specified according to the list at the beginning of Section 4. This is also seen in Figure 5, where a minimization is indicated with a downward-pointing arrow, while a maximization is represented by an upward-pointing arrow.

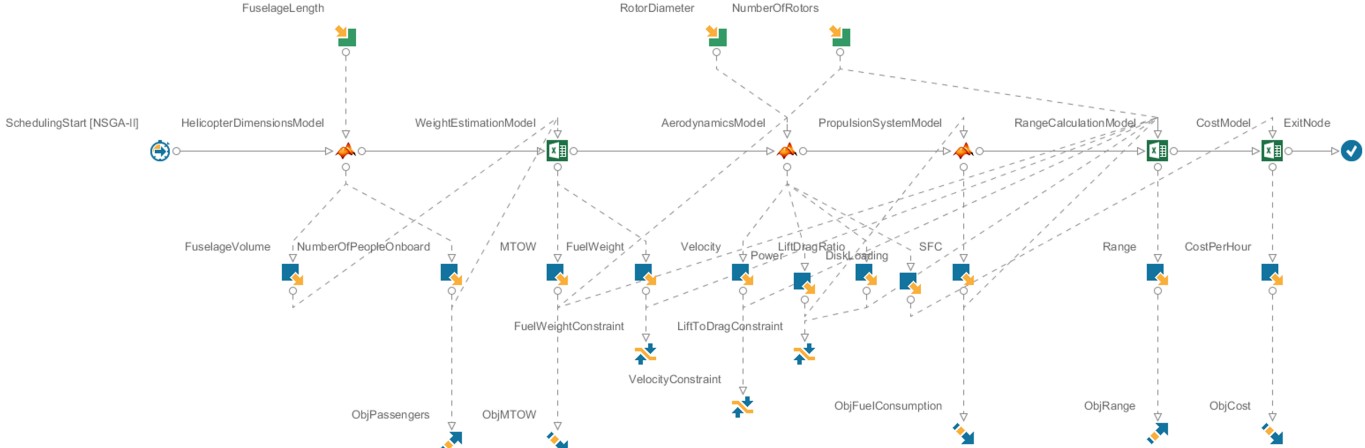

**Figure 5.** The modeFRONTIER optimization framework.

4.2.6. Optimization Setup

As seen in the starting node of the optimization framework illustrated in Figure 5, the evolutionary *NSGA-II* algorithm was chosen for the subsequent optimization. The number of generations for the *NSGA-II* algorithm was set to 150. All other *NSGA-II* settings were left at the default modeFRONTIER values. A design of experiments (DOE) was also specified in order to initialize the optimization and provide a first population. In this particular case study, the *uniform Latin hypercube* algorithm was used to provide a DOE with 20 different designs. This DOE is presented in Table 4 below. Consequently, 3000 evaluations were to be performed during the optimization according to the DOE and *NSGA-II* generations.

*4.3. Results*

This section presents the optimization results that were obtained from the case study and optimization framework outlined above. It shows the available design space and also identifies the Pareto fronts between some of the different optimization objectives. Figure 6 shows the optimization results for the two most important objectives for this particular search and rescue (SAR) case study; namely, the number of people onboard and the range that the helicopter is able to fly.

As seen in Figure 6, two clusters of data points can be identified. This is due to the *number of rotors* design parameter. All one-rotor configurations are found to the left in Figure 6, while the dual-rotor configurations are found more towards the right-hand side. Two Pareto fronts can therefore be identified, as highlighted in blue in the figure. It can also be seen that the dual-rotor configurations generally have a greater range and passenger capacity than the single-rotor configurations. However, Figure 6 only illustrates two of the five optimization objectives, and further evaluations are therefore required in order to draw a conclusion about the most suitable design for SAR operations in this case.

There are various ways of illustrating all the design parameters and optimization objectives simultaneously. A sensitivity matrix can give insight into how the different design parameters affect all the optimization objectives. As the name implies, it is also possible to see how sensitive the different objectives are to the design parameters. Figure 7

presents a sensitivity matrix, with the design parameters on the horizontal axis and the optimization objectives on the vertical axis.

**Table 4.** The utilized design of experiments (DOE) with its 20 different designs.

| ID | Category | Fuselage Length [m] | Number of Rotors [-] | Rotor Diameter [m] |
|----|----------|---------------------|----------------------|---------------------|
| 0 | ULH | 13.678 | 2 | 20.066 |
| 1 | ULH | 39.668 | 1 | 4.835 |
| 2 | ULH | 7.581 | 1 | 13.182 |
| 3 | ULH | 37.910 | 1 | 15.783 |
| 4 | ULH | 12.680 | 1 | 5.864 |
| 5 | ULH | 19.227 | 2 | 21.868 |
| 6 | ULH | 4.470 | 2 | 8.406 |
| 7 | ULH | 28.758 | 1 | 24.122 |
| 8 | ULH | 33.890 | 2 | 12.229 |
| 9 | ULH | 16.208 | 1 | 13.831 |
| 10 | ULH | 35.622 | 2 | 10.291 |
| 11 | ULH | 30.580 | 1 | 19.087 |
| 12 | ULH | 31.198 | 2 | 15.332 |
| 13 | ULH | 27.331 | 2 | 7.275 |
| 14 | ULH | 10.134 | 2 | 6.408 |
| 15 | ULH | 24.421 | 1 | 10.590 |
| 16 | ULH | 23.584 | 1 | 21.315 |
| 17 | ULH | 8.225 | 1 | 17.664 |
| 18 | ULH | 20.228 | 2 | 17.489 |
| 19 | ULH | 18.028 | 2 | 23.204 |

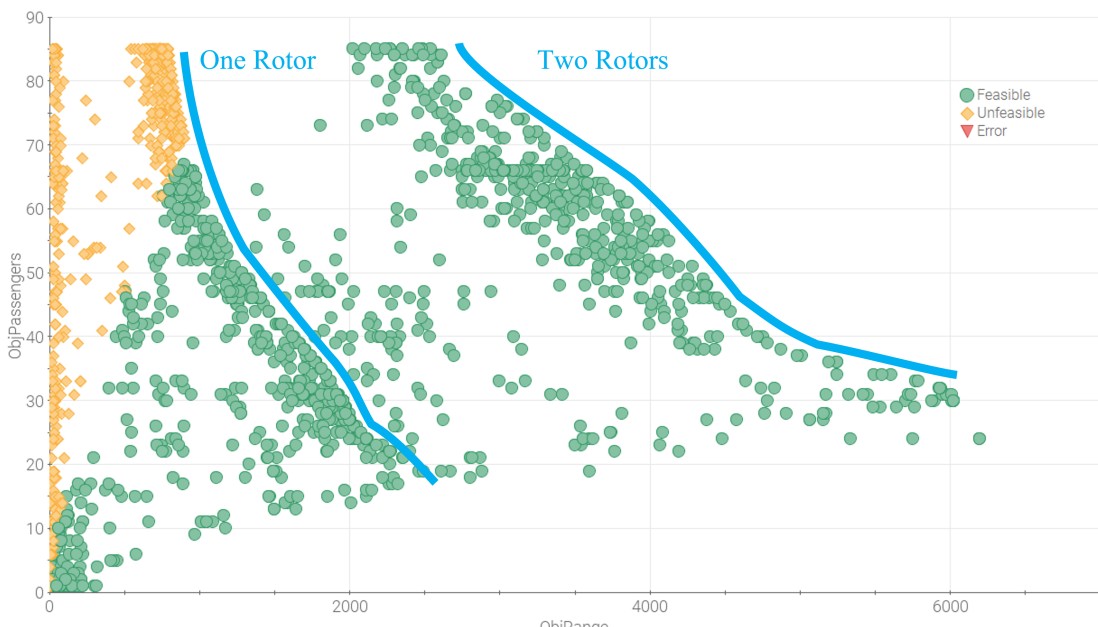

**Figure 6.** A scatter graph for the number of passengers against the helicopter range. Two Pareto fronts for the number of rotors are identified and marked in blue.

As seen in Figure 7, the fuselage length is closely correlated with the range, passenger capacity, and MTOW. An increase in fuselage length would consequently result in an increase in these objectives as well. This is reasonable since the passenger capacity and

weight are both directly tied to the size of the helicopter. The increase in range with a longer fuselage is probably due to the greater possible fuel capacity in a larger helicopter.

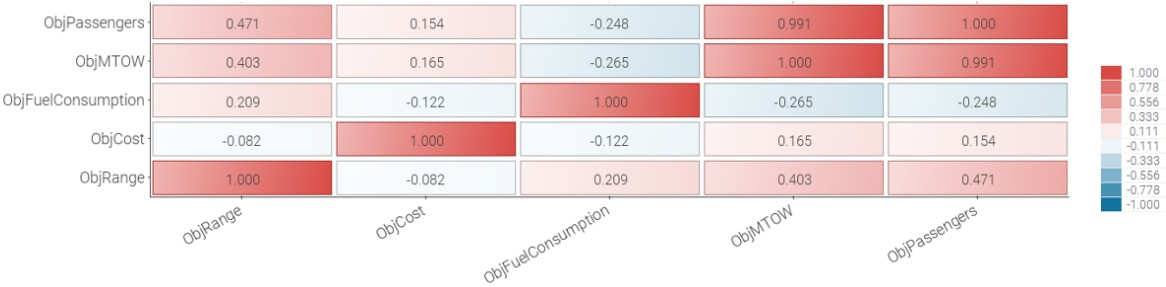

**Figure 7.** A sensitivity matrix between design parameters and optimization objectives.

The correlations between objectives can be illustrated in a system characteristic correlation matrix. Figure 8 illustrates such a matrix where all optimization objectives are correlated against each other.

**Figure 8.** A system characteristics correlation matrix illustrating the relationships between the optimization objectives.

Naturally, the correlation matrix in Figure 8 becomes symmetrical and illustrates both relationships and possible design conflicts between the used objectives. Here, a value of 1 indicates that two objectives are completely in line and that an increase in one of them would yield a similar response in the other. Negative values in the matrix indicate that the objectives are pulling in opposite directions. Values close to zero indicate weak correlations between objectives, while a value of zero would correspond to no correlation at all. For example, the MTOW objective in Figure 8 has a strong positive correlation to the amount of passengers. Cost and range are also positively correlated and would consequently rise with increased MTOW. However, the fuel consumption objective has a slight negative correlation and would therefore decrease as MTOW increases.

These observations from the matrices in Figures 7 and 8 can also be seen in, for example, a parallel coordinates graph. Consequently, Figure 9 presents such a graph, where all design parameters and optimization objectives have been added.

Each line in Figure 9 represents one specific design. It can also be seen here that the fuselage length, number of passengers, and MTOW more or less follow each other (as also indicated in Figures 7 and 8). Consequently, the MTOW objective can be regarded as closely coupled to the number of passengers, thus "reducing" the overall number of objectives to four. As a result, a four-dimensional bubble chart is finally presented in order to identify the most suitable solutions to this specific SAR case study. This bubble chart can be seen in Figure 10.

As Figure 10 shows, the dual-rotor configurations are superior from a range and passenger capacity point of view. However, these generally have a higher fuel consumption for the same number of passengers compared to the single-rotor helicopters. Another interesting observation is that high-capacity, single-rotor helicopter configurations are typically more expensive to operate than their dual-rotor counterparts. A possible explanation for this could be the required power and size of the corresponding engines per rotor. With all remaining optimization objectives taken into consideration, the most suitable design

region is likely to be on the "Pareto front" for the dual-rotor configurations, with a range of around 4500 km and a capacity of about 50 passengers. This region is marked with a black ellipse in Figure 10.

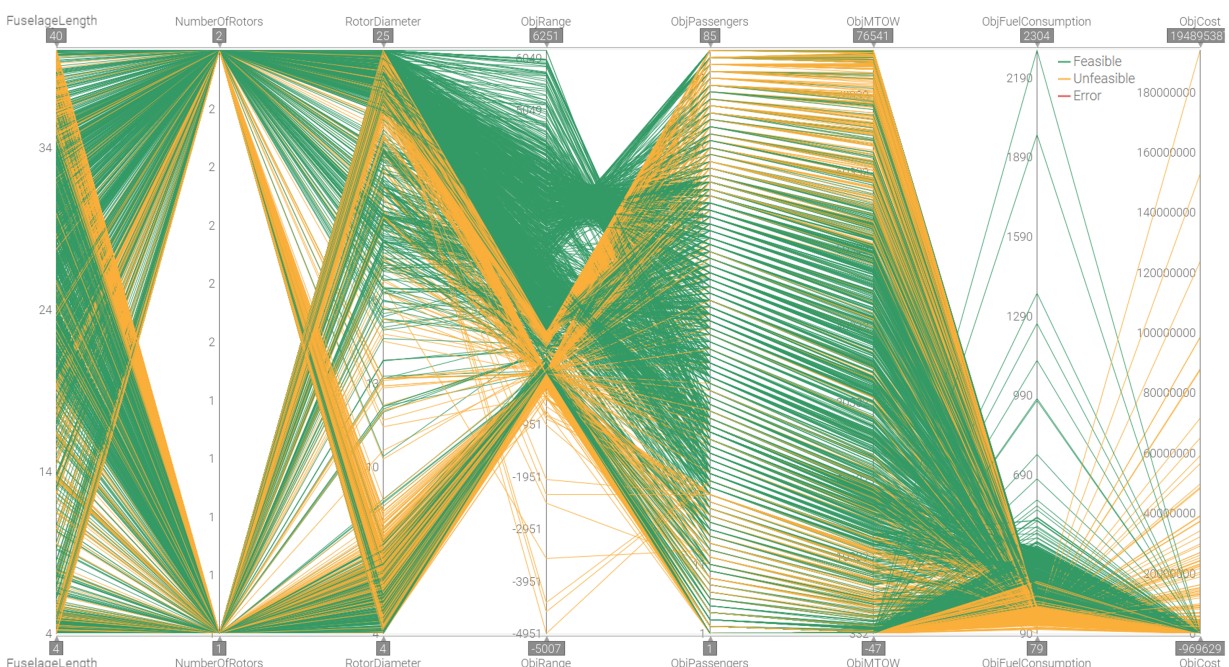

**Figure 9.** A parallel coordinates graph showing the relationships between design parameters and optimization objectives for each design alternative.

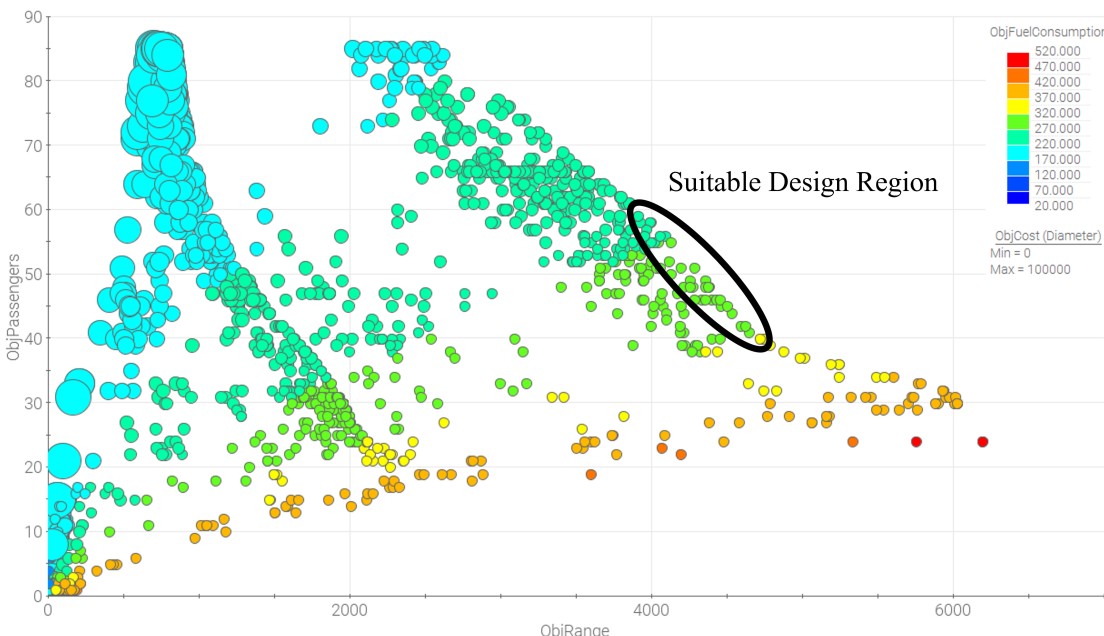

**Figure 10.** A four-dimensional bubble chart, where the range objective is represented on the x-axis, the passenger capacity on the y-axis, the fuel consumption as the color gradient, and the overall cost objective as the size of the individual bubbles.

## 5. Discussion and Future Work

The results of this case study show how the method and workflow illustrated in Figure 3 can be used to create an optimization framework for early design studies. More specifically, the results of the case study were used to propose a helicopter design configuration for SAR missions. It should be noted, however, that the optimization objectives for

this case study were set for a general-purpose mission, and in reality it might not be very efficient to send a 50-passenger helicopter to search for, and rescue, a single survivor lost at sea. No particular solution has been chosen at this point, and consequently the design space is kept open for possible changes in boundary conditions and optimization objectives. This gives decision-makers an available trade space, where different objectives can be traded against each other in an investigation of different possible future SAR scenarios. Additionally, this is similar to the practice of *"set-based design"*, where no single point solutions are chosen until the very end of the design process, or when sufficient knowledge about the available design space has been obtained [32].

The case study performed here was relatively simple in terms of the design parameters involved and models used. However, the intention in choosing an MDO framework with different models was to keep it modular. This means that the models used can be exchanged for higher-fidelity ones if desired. As an example, in the future, the aerodynamics model could be exchanged for a more detailed computational fluid dynamics (CFD) simulation. This may give more accurate results for the overall framework. More design parameters could also be added in the future. The size and number of engines used is one of the most important future additions to the SAR helicopter case study. There are also many other details that could be taken into consideration to provide a more detailed SAR helicopter case study overall. Resistance to environmental conditions, such as weather and wind, are important parameters that were not included in the case study presented here. As another example, the coefficient of power loss and rotor efficiency factor in the range equation were considered as constants in the case study. These could be determined based on, for example, flight velocity in a future more detailed study. However, this case study does provide relatively realistic results despite the simplifications and lack of high-fidelity models. Consequently, the case study proves that "ballpark" estimates can be obtained for a design with relatively little detail and at "low" computational cost.

As mentioned in Section 3, the proposed method and workflow for creating an optimization framework intended for early design studies was kept domain-neutral. The workflow in Figure 3 is consequently intended to be applicable to other designs as well, given that underlying relationships can be determined; for example, by using statistics. However, its applicability to domains other than SAR helicopters remains to be tested. More regression analyses using symbolic regression and optimization are topics for future work. These regression analyses are also domain-neutral and work on any underlying data. However, expert knowledge from the intended domain can facilitate the creation of regression models. For example, it might not be obvious to a non-helicopter design expert that a helicopter disk loading is best correlated against fuel consumption and power. Certain improvements in the existing regression models for the case study might therefore be in order. As most of the relationships for the case study in this paper were based on statistics taken from existing solutions, evaluations of more novel helicopter design solutions might be subject to significant uncertainties. Consequently, more analytical calculation models should be included in that case. For example, at the moment, the velocity constraint used in the case study limits the design space. In reality, workarounds could be performed to achieve higher cruising speeds without stalling the rotor blades. However, this will not be captured by the statistical analysis if no such designs are included. Nevertheless, the overall purpose of the case study was to illustrate how the method and workflow illustrated in Figure 3 can be used to obtain an optimization framework for early design studies. The level of detail in the corresponding SAR helicopter case study is therefore regarded as only one of many possible avenues for future work.

*Outlook*

One of the most prominent avenues for future work on the overall optimization framework is to investigate the creation of surrogate models from the obtained case study results. Surrogate models would allow for fewer evaluations during optimizations, but would also provide "cheap" estimation models from a computational cost perspective. A possible

*surrogate model definition* step is planned, and will be added to the method and workflow presented in Figure 3 in the future. In that case, the outcome of the workflow would not be a single optimum design for the intended domain, but rather a framework intended for design and trade space explorations. As mentioned previously, such a framework could aid in aspects such as decision support, "what-if" investigations, and forecast analyses from possible future scenario perspectives.

Another topic for future work is to further investigate the results obtained from the optimization framework using visual analytics (VA). VA is used to create interactive visualizations from underlying data that can be used to identify hidden relationships and facilitate decision support [33]. It also supports human reasoning in the analysis of very large datasets. Consequently, an interactive dashboard with different visualization and filtering techniques can be employed to give users an intuitive overview of the entire optimization process. Such an interactive dashboard could also be used in that case as a design platform where, for example, models can be interactively exchanged for more detailed ones to determine how the overall optimization results are affected. Underlying surrogate models would be very beneficial from a computational cost perspective in this case as well.

## 6. Conclusions

This work shows how an optimization framework for early design studies can be created using a proposed method and workflow. A case study for the design of a general-purpose search and rescue (SAR) helicopter was introduced and used to illustrate this intended workflow. Analytical relationships, as well as various regression models taken from statistics on existing helicopters, were used to provide different models for each helicopter design discipline. The generated models were then interconnected in a multidisciplinary design optimization (MDO) framework, together with available design parameters, optimization objectives, and constraints. The subsequent optimization resulted in a design space of different solutions in which a Pareto front of suitable designs for the general-purpose SAR helicopter could be identified. Overall, this work illustrates how optimizations and relatively few details about a domain can be used to make fairly accurate "ballpark" estimates, which are usually enough at an early conceptual design stage.

**Author Contributions:** Conceptualization, L.K.F., I.S., P.K. and K.A.; methodology, L.K.F. and I.S.; software, L.K.F.; validation, L.K.F., I.S., P.K. and K.A.; formal analysis, L.K.F.; investigation, L.K.F.; resources, L.K.F.; data curation, L.K.F.; writing—original draft preparation, L.K.F.; writing—review and editing, L.K.F., I.S., P.K. and K.A.; visualization, L.K.F.; supervision, I.S., P.K. and K.A.; project administration, P.K.; funding acquisition, P.K. All authors have read and agreed to the published version of the manuscript.

**Funding:** This research was funded by the Swedish Innovation Agency (VINNOVA) under grant no. NFFP7/2017-04838.

**Institutional Review Board Statement:** Not applicable.

**Informed Consent Statement:** Not applicable.

**Data Availability Statement:** The helicopter dataset that was collected and used for the regression analyses in this paper is available from https://gitlab.liu.se/ludfr93/s2tep-ontologies-and-materials (accessed on 6 October 2022).

**Acknowledgments:** The authors would like to acknowledge the contributions of Johan Persson and Fredric Malm at Linköping University. Johan assisted in the performance of the optimization work and reviewed an early draft of this paper. Fredric Malm contributed with expert knowledge from a helicopter design point of view, which was very helpful in the presented case study. Fredric also reviewed a draft of this written work.

**Conflicts of Interest:** The authors declare no conflict of interest.

## Abbreviations

The following abbreviations are used in this manuscript:

| | |
|---|---|
| CFD | Computational fluid dynamics |
| DOE | Design of experiments |
| DSM | Design structure matrix |
| MDO | Multidisciplinary design optimization |
| MOO | Multiobjective optimization |
| MTOW | Maximum takeoff weight |
| SAR | Search and rescue |
| SFC | Specific fuel consumption |
| VA | Visual analytics |

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
