# Peer review of "Optimization Framework for Early Conceptual Design of Helicopters"

_aerospace, doi:10.3390/aerospace9100598_

Round 1

Reviewer 1 Report

Comments on Title:

(1) The title is very general considering the content of the paper. The authors say the framework can be applied to different engineering domains, e.g. helicopters, cars or toasters. However, this paper is only for helicopters. The authors can either make the title more specific or add another example in engineering to demonstrate the generalization of the framework.

Comments on Abstract:

(1) Do not use abbreviation in the abstract

Comments on Introduction:

(1) Generally, there is a lack of literature review on the other optimization frameworks or the helicopter conceptual design methods.

Comments on Approaches and Design Methodologies

(1) Section 2.1 provides a method that is very specific to aircraft/helicopters. This is clearly not a general method for different engineering domains.

(2) The Breguet range equation is more appropriate to fixed-wing aircraft. For the range of helicopters, there are some other factors that need to be considered, such as the rotor efficiency and the coefficiient of power loss.

(3) Section 2.4, Line 165: The design process for fixed-wing aircraft and helicopters can be very different. For fixed-wing aircraft, the design process starts with thrust to weigth ratio vs wing loading, which are not applicable to helicopters.

Comments on Method:

(1) Use present tense. For example, line 183: It is better to use "is" rather than "was".

Comments on Case Study and Implementation:

(1) Line 231: what is the cost model?

(2) There are five objectives and three design variables, which means there are not enough degree of freedom to explore all the objectives.

(3) Section 4.2: it is not clear to the reviewer what are the models for the aerodynamics, propulsion and cost.

Comments on References:

(1) References 18 and 19 need to be formatted.

Author Response

Dear Reviewer 1, thank you for reviewing our manuscript and giving us the opportunity to improve it. The suggested improvements have been considered and added to the manuscript. Among many other things, more literature relevant to MDO and helicopter design have been added to the introduction. The title and overall manuscript have also been adjusted to further emphasize that the work only deals with helicopters. The Breguet range equation has been updated with the suggested additions and the optimizations have been run again. Consequently, all results have been updated. Finally, more details have been added for the individual models in the Case Study and Implementation chapter to make it more transparent for the reader.

For a detailed description of the made changes and adjustments to the manuscript, please see the attached Response to Reviewer 1 Comments.pdf.

Reviewer 2 Report

The paper is in general very well written and only few minor improvements are suggested:

(1) In Introduction, Helicopter design is such an example and is a multidisciplinary domain that often includes contradictory requirements and negotiations to arrive at suitable concept. After this statement, please give an example to support this statement for the ease of understanding of readers.

(2) The purpose of this paper is the same as the early development of MDO theory and it is very strange that the authors did not mention the work in that area in Introduction. It would be better to make some comparisons to justify the present work. 

In overall, I think the paper contains the original contribution for the SAS helicopter and the writing style is clear.  

Author Response

Dear Reviewer 2, thank you for reviewing our manuscript and giving us the opportunity to improve it. The introductory chapter has been updated with examples for possible conflicts in helicopter design and more MDO theory references have been added according to the given suggestions. The additional comments in the supplied .pdf document has also been taken into consideration.

For a detailed description of the made changes and adjustments to the manuscript, please see the attached Response to Reviewer 2 Comments.pdf.
